# Spatiotemporal Patterns of Air Pollution in an Industrialised City—A Case Study of Ust-Kamenogorsk, Kazakhstan

**Daulet Assanov** [1], **Ivan Radelyuk** [2,*], **Olessya Perederiy** [3], **Stanislav Galkin** [3], **Gulira Maratova** [3], **Valeriy Zapasnyi** [3] and **Jiří Jaromír Klemeš** [4]

1   Excellence Center Veritas, D. Serikbayev East Kazakhstan Technical University, Ust-Kamenogorsk 070000, Kazakhstan
2   Department of Chemistry and Chemical Technologies, Toraighyrov University, Pavlodar 140000, Kazakhstan
3   Laboratory "Atmosphere", D. Serikbayev East Kazakhstan Technical University, Ust-Kamenogorsk 070000, Kazakhstan
4   Sustainable Process Integration Laboratory—SPIL, NETME Centre, Faculty of Mechanical Engineering, Brno University of Technology—VUT, 616 56 Brno, Czech Republic
*   Correspondence: radelyuk.i@tou.edu.kz

**Abstract:** Air quality issues still affect the quality of life for people in industrialised cities around the world. The investigations should include the identification of the sources of the pollution and its distribution in space and time. This work is the first attempt to perform identification of the sources of pollution in Ust-Kamenogorsk city in Kazakhstan. Analysis of retrospective data (including ten variables (TSP, $SO_2$, CO, $NO_2$, phenol, HF, HCl, $H_2SO_4$, formaldehyde, $H_2S$) from five monitoring stations for the period 2017–2021) using multivariate statistical methods and hierarchical cluster analysis has been performed to assess spatiotemporal patterns of air quality of the city. The results indicate that the contamination patterns can be grouped into two categories: cold and warm seasons. The study revealed the dangerous concentrations of $NO_2$ and $SO_2$ exceeded the limits by 2–3 and 1.5–2 times, independently of the seasonality. Averaged concentrations of TSP slightly exceeded the established limits for the most industrialised part of the city. Concentrations of HF and formaldehyde significantly rose during the cold seasons compared to the warm seasons. Other chemical parameters significantly depend on the seasonality and locations of the sampling points. The major reason for air pollution is twofold—the use of a burnt-coal throughout the year for electricity and heat generation (especially during the cold seasons) and the high density of the heavy metallurgy industry in the city. The principal component analysis confirms a high loading of industrial sources of air pollution on both spatial and seasonal dimensions.

**Keywords:** air pollution; air quality; cluster analysis; industrial emissions; principal component analysis

## 1. Introduction

In 2022 humanity still faces air pollution issues, while it is much known about efficient ways to handle them. Statistics revealed the significant pollution in big cities, even in countries where the possibility for permanent monitoring with the wide web of monitoring stations has been established [1]. At the same time, the situation can be much worse in developing countries, especially in industrialised cities, as the cities have not been connected to the common database and the ability to assess environmental and social consequences has been hidden. Studies say that despite direct hazards to health and the economy, the influence and the interests of big manufacturing and energy companies sometimes can prevail over the mentioned issues and exert pressure on regulatory aspects [2]. These factors make the air pollution issues still actual around the world. For instance, a study by Yuan et al. [3] claims that cities with pollution-intensive industries are responsible for urban air pollution (called "super emitters" as they dominate as large point sources of pollution [4]). According to another study by Gu et al. [5], the industrial sector has been

recognised as the largest contributor to air pollution among all sectoral emitters in China. Industrial activities led to accounting for 36% of the total impact on health, including premature mortalities, as well as 41% of crop production loss over the whole country. The authors estimated that the scenario that mainly reduces industrial emissions would provide the largest air quality benefits among various scenarios. A sound work by Burnett et al. says that 8.9 million deaths, even as early as in the year 2015, could have been caused by poor air quality worldwide [6]. An SDG's Indicator 3.9.1 states that the mortality rate attributed to household and ambient air pollution should be substantially reduced by 2030 to ensure healthy lives and promote well-being for all.

Kazakhstan is a developing country which sets a collection of the mentioned problems in the context of air pollution. Obsolete and non-conventional equipment in the main emitters [7], especially in the metallurgy sector [8], legislative loopholes [9], denying or slowing implementation of clean technologies [10], and drawbacks in the monitoring systems [11] have brought Kazakhstan to the list of countries with severe air conditions [12]. While Kazakhstan formally aims to achieve SDGs, including a healthy environment and good health for all, there is a lot to do, including air quality problems [13].

One of the problematic cities in the country, Ust-Kamenogorsk, is located in a major mining and smelting area and has been rated among the worst cities regarding the problems of ecology according to World Bank [14]. Extreme pollution events from accidental emissions are not rare in the city [15]. Descriptive studies of poor air quality in the city have said that the city has been rated the most $SO_2$-polluted city in Kazakhstan and among the five most-polluted $NO_2$ and $O_3$ cities in Kazakhstan from 2011 to 2017 [12]. A study about the impact of COVID-19 lockdown on air quality in the city has revealed that the level of CO has decreased by 21–23% with the increase in the TSP level by 13–21%, and had no significant effect on $SO_2$ and $NO_2$ concentrations in the city [16]. Elevated levels of trace elements, particularly Ba, Mn, Pb, V, and Zn, in the blood of city residents of Ust-Kamenogorsk have been found [17], which could indicate a severe impact on the industrial activities of the city. The known studies are limited by their descriptive origin, while there is a demand for deep investigations of the relationships between a high level of air pollution, weak environmental regulation, pyrometallurgical processes and dependence on coal-burnt energy.

Air quality plans apply the following measures for emission reduction: policy implications, control for technical feasibility, calculating resulting costs, source apportionment techniques, and assessment of the impact on the environment and human health. All these measures are based on the assurance of establishing and monitoring the exceedances of the air quality standards [18]. Also, a comprehensive emission inventory coupled with air quality monitoring is the first step to developing an emission control strategy for selected pollutants [19]. These tools do not work on their own, and pollution treatment programs have to come forefront [20]. Stricter environmental regulation leads to a reduction in polluting industrial activities on the way to the green economy and using environmentally-friendly advanced technologies [21]. As a result, embedding advanced control technologies within sustainable development policy scenarios would result in significant cost savings [22]. However, it also can lead to the closure of emitters or their relocation, which negatively impacts economic growth [23].

According to an opinion from [24], preliminary investigations play a core role in the identification of the sources of pollution to drive "the wheel of progress: Sources ⇒ Effects ⇒ Regulation ⇒ Control". There are known different ways to investigate potential sources apportionment: emission inventories [25], inverse modelling [26], artificial neural networks [27], receptor modelling methods [28], air quality models [25], a combination of the different approaches and models [29], application of positive matrix factorisation [30], and even chemical analysis of biomonitoring species [31]. However, all of the listed methods demand a very detailed dataset with a number of monitored parameters. This limitation is a serious obstacle in the conditions of many developing countries, including Kazakhstan,

where the system of permanent monitoring with extended parameters, such as PMs, has still been established [32].

Under these conditions, one of the efficient ways to investigate sources apportionment with the following understanding of the needed steps for air quality management can be multivariate statistical techniques, particularly Principal Component Analysis (PCA) and Cluster Analysis. In addition, using analysis of scores and loadings in PCA can give a representation of potential chemical reactions in ambient air [33]. The approach to applying PCA for the assessment of patterns in air quality has already been applied several times. For instance, Dominick et al. [34] aimed to investigate possible sources of air pollutants and spatial patterns in Malaysia using the same techniques as the authors plan to do with grouping particulate contaminants in principal components. Application of the PCA to find the correlation between gaseous pollutant concentrations, meteorological factors and potential sources of pollution has identified the contribution of combustion- and non-combustion-related emitters in Greece [35] or in India [36]. The tool has been used for the same purpose also by Azid et al. in the context of the prediction of air pollution [37]. PCA has supported the assessment and identification of the sources of air pollution in India, where complex industrial activities exist [38]. Revealing information about the sources and mechanisms of air pollution in Madrid and visualising their spatial distribution using PCA and the geostatistical method has been carried out by Núñez-Alonso et al. [39]. Particular attention has been paid to the investigation of the presence of metals only in particulate matters coupled with multivariate statistical techniques in an Iranian industrial city [40]. For example, a combination of the analysis of the chemical composition of fine particulate matters with the focus on metals presence with Absolute Principal Component Analysis allowed attributing the identified pollutants to their sources in the USA [41].

The aim of this study is to analyse key factors impacting air quality in Ust-Kamenogorsk using the available dataset for the period 2017–2021 by multivariate statistical techniques on spatial and temporal scales. This approach enables us to investigate potential sources of apportionment using the large but limited dataset of the observations in the city for the first time.

## 2. Materials and Methods

### 2.1. Study Area

Ust-Kamenogorsk (or Oskemen) is located in northeastern Kazakhstan in the foothills of the Altay and at a confluence of the Irtysh and the Ulba rivers. The climate of the Ust-Kamenogorsk region is temperate continental. The city is divided into two parts by the Irtysh River. The city is surrounded by Shanovsky and Kalbinsky mountain ranges on the southeastern site [42].

A number of the largest Kazakhstani metallurgy plants for the production of non-ferrous metals are located in the city: the metallurgical complex of Kazzinc LLP, the Ulba metallurgical plant, and the Ust-Kamenogorsk titanium and magnesium plant. Coupled with the Ust-Kamenogorsk and the Sogrinskaya thermal power plants, the industrial activities put significant pressure on the air conditions of the city. The locations of the main industries are presented in Figure 1. The main characteristics of the industries of the city are presented in Table 1.

The city can be conditionally divided into three zones: two big industrial areas (the northern and the northeastern industrial zones), and downtown, located on the left bank of the Irtysh River, with its own thermal power plant. The northern industrial zone includes the locations of the Ust-Kamenogorsk metallurgical complex of Kazzinc LLP, the Ulba metallurgical plant, and the Ust-Kamenogorsk thermal power plant. The northeastern industrial zone includes the Ust-Kamenogorsk titanium-magnesium plant and Sogrinskaya thermal power station. Accordingly, five monitoring stations (noted as S1–S5 in Figure 1) are located for sampling and analysis of the air quality:

- Station 1—is located in the northern industrial zone;
- Station 2—is located in the administrative centre of the city;

- Station 3—is located in the north-western part of the city, adjacent to the northern industrial zone;
- Station 4—is located in the northeastern industrial zone;
- Station 5—is located downtown.

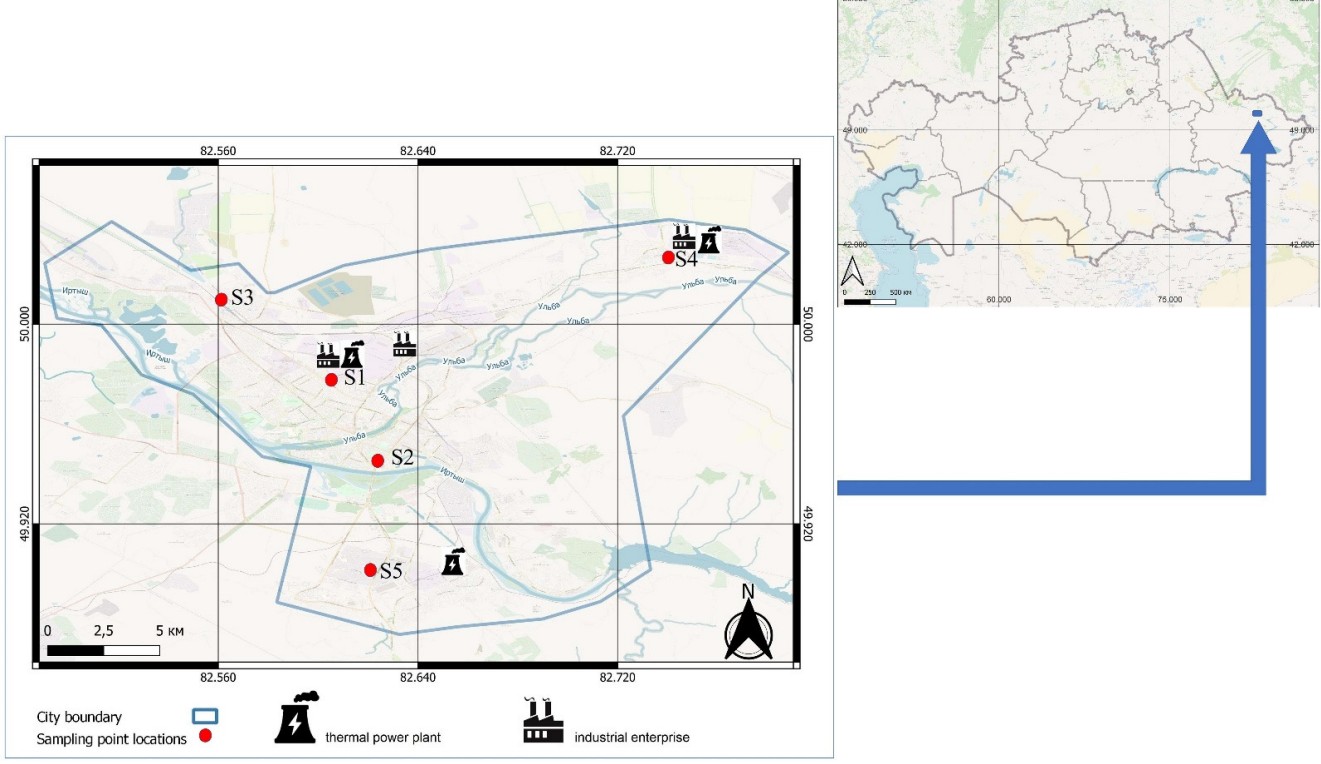

**Figure 1.** Map of the study area.

**Table 1.** Characteristics of the industrial enterprises of Ust-Kamenogorsk.

| Enterprise | Production | The Volume of the Production | Permitted Emissions, t/y |
|---|---|---|---|
| The Ust-Kamenogorsk metallurgy complex of Kazzinc LLP | Lead<br>Zinc<br>Copper<br>Technical sulfuric acid | 144 Kt/y<br>190 Kt/y<br>70 Kt/y<br>1000 Kt/y | TSP—129;<br>$NO_2$—245;<br>$SO_2$—16,856;<br>$H_2SO_4$—51;<br>CO—8356;<br>HF—12;<br>HCl—57;<br>$H_2S$—0.5 |
| The Ulba metallurgical plant | Tantalum<br>Uranus<br>Beryllium | No open data | TSP—7.4;<br>$NO_2$—5;<br>$SO_2$—0.1;<br>$H_2SO_4$—4.4;<br>CO—0.5;<br>HF—2.3;<br>HCl—0.7 |
| The Ust-Kamenogorsk thermal power plant | Heat energy<br>Electricity<br>Burnt coal | 859.9 Gcal/h<br>372.5 MW/y<br>1.5 Mt/y | TSP—3035;<br>$NO_2$—4470;<br>$SO_2$—9277;<br>CO—185;<br>HF—0.006;<br>$H_2S$—0.001 |

**Table 1.** *Cont.*

| Enterprise | Production | The Volume of the Production | Permitted Emissions, t/y |
|---|---|---|---|
| The Ust-Kamenogorsk titanium and magnesium plant | Titanium tetrachloride<br>Sponge titanium<br>Raw magnesium<br>Anhydrous carnallite | 49 Kt/y<br>12 Kt/y<br>13 Kt/y<br>9.6 Kt/y | TSP—24;<br>$NO_2$—15;<br>$SO_2$—30;<br>$H_2SO_4$—0.1;<br>CO—314;<br>HF—0.14;<br>HCl—28 |
| The Sogrinskaya thermal power plant | Heat energy<br>Electricity<br>Burnt coal | 168 Gcal/h<br>75 MW/y<br>0.35 Mt/y | TSP—454;<br>$NO_2$—975;<br>$SO_2$—1974;<br>CO—8 |
| The left-bank thermal power plant | Heat energy<br>Burnt coal | 168 Gcal/h<br>0.083 Mt/y | TSP—693;<br>$NO_2$—342;<br>$SO_2$—446;<br>CO—95 |

### 2.2. Multivariate Statistical Techniques

Correlation analysis, principal components analysis (factor analysis), and hierarchical cluster analysis were applied to identify the multivariate relationships between different variables and samples in the study area. The dataset was normalised for the elimination of the effect from differences in units (Equation (1)).

$$Z_{ij} = \frac{(x_{ij} - m_i)}{SD}, \tag{1}$$

where $Z_{ij}$ are normalised values from $x_{ij}$, $i$ is the represented variables, $j$ is the sample number, $m_i$ is the mean value, and SD is the standard deviation of the sample.

The relation between each pair of variables was measured by Pearson's correlation coefficient to determine the associations among different variables. Correlation coefficients greater than 0.5 were considered significant. PCA recognises the most significant parameters from a big dataset of inter-correlated parameters and creates independent variables (Equation (2)).

$$z_{ij} = a_{i1}x_{1j} + a_{i2}x_{2j} + \ldots + a_{im}x_{mj}, \tag{2}$$

where $z$ is the component score, $a$ is the component loading, $x$ is the measured value of a variable, $i$ is the component number, $j$ is the sample number, and $m$ is the total number of variables. Factor analysis (FA) is a similar approach to PCA. However, PC is presented as a linear combination of parameters. FA follows PCA and takes into account unobservable, hypothetical, latent variables. They are included in the equation with the special residual term (Equation (3)).

$$z_{ij} = a_{f1}f_{1j} + a_{f2}f_{2j} + \ldots + a_{fm}x_{mj} + e_{fi}, \tag{3}$$

where $z$ is the measured variable, $a$ is the factor loading, $f$ is the factor score, $e$ is the residual term according to errors or another source of variation, $i$ is the sample number, and $m$ is the total number of factors.

Cluster analysis was used to assemble similar groups of the monitoring dates due to similarities between their variables. Hierarchical agglomerative CA provided Ward's linkage distance, reported as $D_{link}/D_{max}$, which represents the quotient between the linkage distances for each case divided by the maximal linkage distance. The produced dendrogram enables analysing similarities easily. Ward's linkage and the Euclidean distance as similarity measurements are commonly used for cluster analysis for the assessment of air quality [43].

All mathematical and statistical computations were performed using Microsoft Office Excel 2016 and IBM SPSS Statistics 26 software.

*2.3. Data Management and Methodological Framework*

The National Hydrometeorological Service of Kazakhstan "Kazhydromet" has provided a raw dataset for this study. Equipment, lab, staff, and methodology of analysis are certified and follow national and international standards for QC/QA. The dataset contains the results of manual measurements of 17 contaminants. Most of these measurements have been carried out four times per day at five stations (Figure 1). To obtain the most detailed picture, ten contaminants have been chosen for spatiotemporal and statistical assessment: total suspended particles (TSP); $SO_2$; CO; $NO_2$; phenol; HF; HCl; $H_2SO_4$; formaldehyde; and $H_2S$ for the period 2017–2021. The reason for the selection of these contaminants and this period is the fullest available dataset, as they have been measured on a regular daily basis, except for weekends and vacations. It is important to note that TSP is not a commonly used parameter worldwide, while information about common worldwide contaminants particulate matter (PM) has not been provided due to its absence. Information about temperature, wind speed, wind direction, and humidity has been obtained from the archive of the web resource www.rp5.kz (accessed on 15 November 2022) [44] to identify possible interconnections with meteorological conditions of the region.

The first step of the analysis was to identify potentially similar periods in a matter of contamination. This step includes hierarchical clustering analysis, which was performed for each studied year separately with the following evaluation and identification of the grouped periods. Dates against daily averaged values of contaminants have been used as parameters for this analysis. The daily averaged values of the contaminants were also used for the second step: performing descriptive statistical analysis to evaluate air pollution in general for the selected temporal clusters. The number of daily averaged observations was 859 and 602 for the cold and warm seasons for all the monitored contaminants. The spatiotemporal assessment was the third step of this study and included time series and spatial analysis using geoinformation systems. Monthly averaged values were used to identify and evaluate trends in air pollution through the studied period, while mean values for the whole period were used for the spatial assessment. Interpolation using the inverse distance method was used to describe the distribution of the contamination within the city [45]. The fourth step of this study was to perform PCA based on the correlation matrix. The correlation matrix was built using daily averaged values of the studied contaminants coupled with the meteorological parameters, while the PCA was completed using chemical parameters only on a daily averaged measurements basis.

## 3. Results and Discussion

*3.1. Hierarchical Clustering Analysis*

Clustering analysis has been performed using dates as variables based on parameters of contamination to identify the similarities within particular periods. The final result of the analysis can be seen in Figure 2. The results indicate that the studied period can be grouped into two categories: cold (including months from September to March) and warm (from April to August) seasons. This can be explained by the fact that contaminants (except formaldehyde) have shown maximum concentrations during cold seasons with peaks during the months of December–February (Figure 3). Thus, the following assessment includes a separate analysis of air pollution patterns in both cold and warm seasons. These findings in seasonality can also be used in better planning for preparing for avoiding public health burdens during the seasons of interest [46].

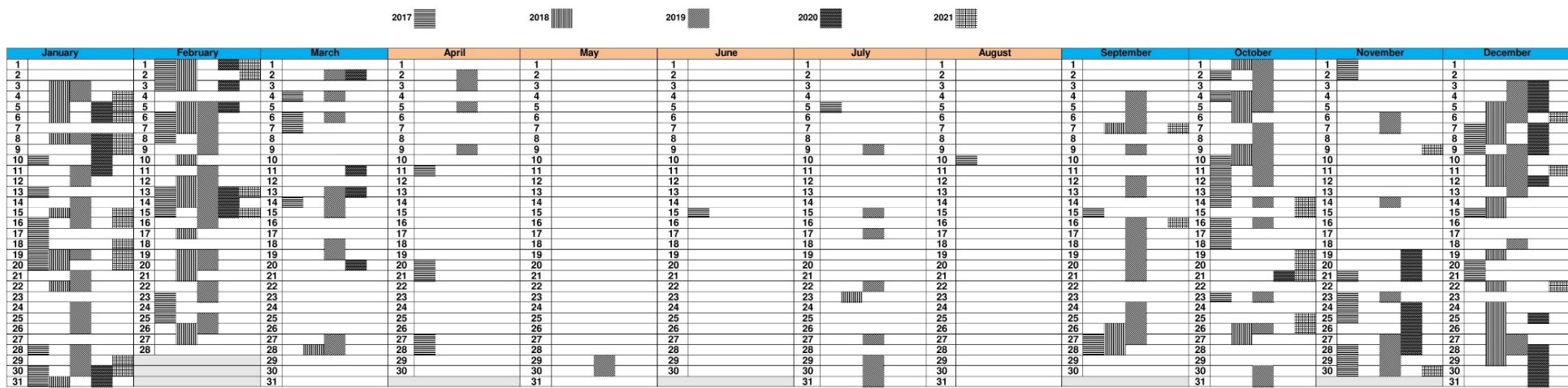

**Figure 2.** Results of the hierarchical clustering analysis.

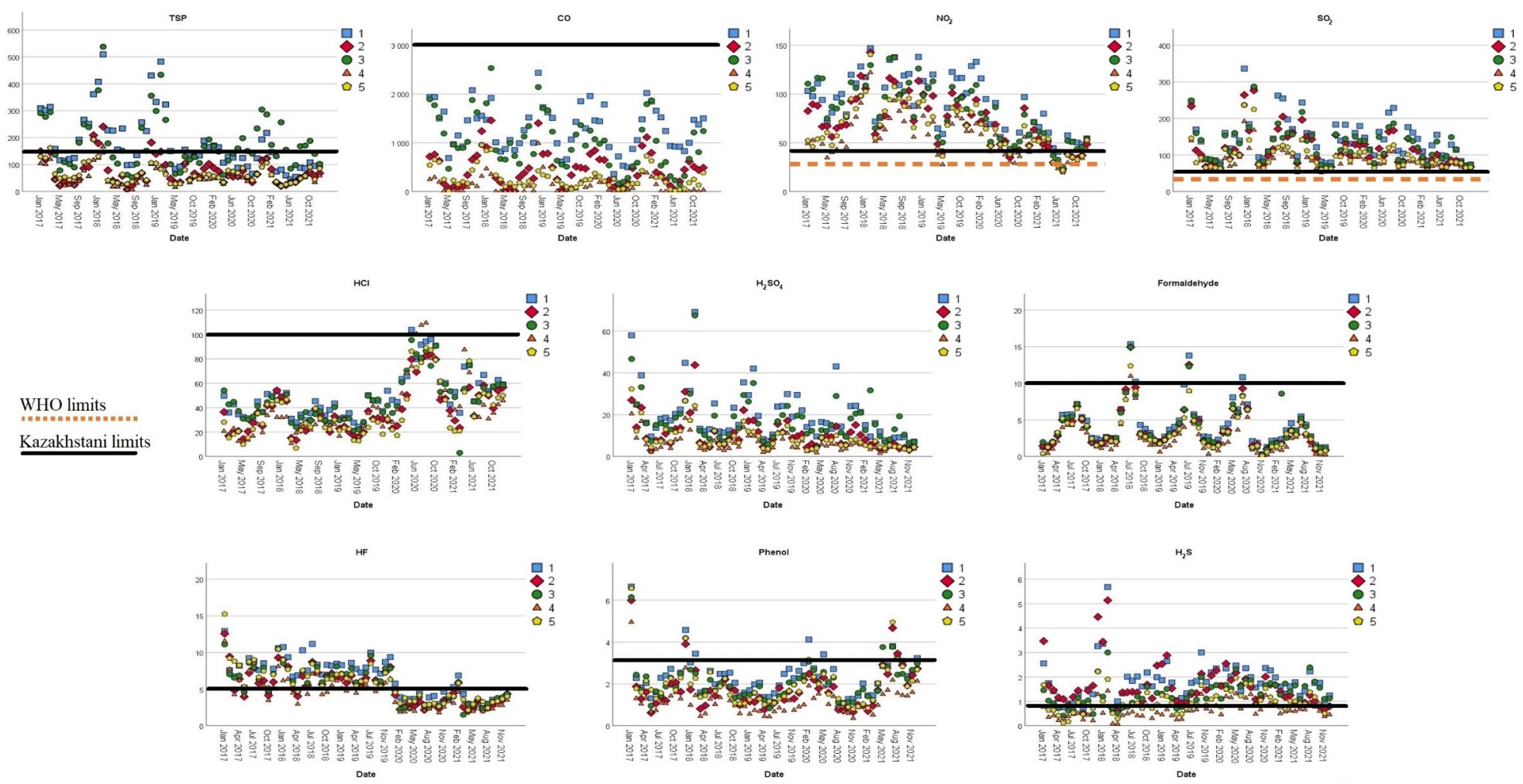

**Figure 3.** Temporal variations of the studied pollutants against WHO and Kazakhstani limits ($\mu$g m$^{-3}$).

### 3.2. Descriptive Statistics

Tables 2 and 3 present the results of measurements of air quality from the monitoring stations in the city according to the limits established by the Kazakhstani government and recommended by the WHO. It is clearly seen that heavy industry and coal-burnt energy cause the permanent exceeding of the permissible daily values along the city during both cold and warm seasons. The worst situation remains for $NO_2$ and $SO_2$, which are significantly higher than both limits (by 2–3 times for $NO_2$ and by 1.5–2 times for $SO_2$) during the whole period of analysis (Figure 3). Averaged concentrations of TSP slightly exceeded the established limits for Stations 1 and 3 during the cold seasons, with the peak values during January months in 2017 and 2018 (Figure 3), while during the warm seasons, they can be characterised as safe. Concentrations of HF and $H_2S$ show slight excess over the recommended concentrations along all stations (except Station 4) during both seasons. However, the presence of HF dropped below the limit line after January 2020 (Figure 3). Surprisingly, the averaged and median concentrations of CO have not shown exceeding values, except several daily exceedings of the parameter were recognised during the cold seasons. In general, the concentrations of the pollutants show a slightly descending trend while they still remain much above the permissible level. It can be explained by the report from the National Statistics Committee, which claims that industrial emissions decreased from 29 to 27.9 kt/y for $SO_2$ and from 10.8 to 10.4 kt/y for CO for the period between 2017 and 2021 [47].

It is fair to note that the recently updated Kazakhstani limits [48] have not followed recommendations from international standards [49] and experts [50]. The limits for main pollutants have not been revised and still are above the recommendations from WHO [51].

The spatial distribution of the contaminants is presented in Figures 4 and 5. Only TSP, $SO_2$, $NO_2$, HF, and formaldehyde show the most significant patterns in their dispersion within the city. It is clearly seen that the major emitter of the city is located near Station 1 and represents the northern industrial zone with two huge metallurgy factories and one thermal power plant. Surprisingly, the safest location in the city regarding concentrations of the pollutants is near Station 4, despite its close location to the northeastern industrial zone. The main difference between seasons is the presence of pollution near Station 5, in the southern direction from the major emitters to the downtown, which is located on the left-bank part of the city with its own thermal power plant. While concentrations of HF and formaldehyde look high, comparatively with the centre of the emissions during the cold seasons, the presence of these contaminants during the warm seasons looks safe and strives for minimal values within the city (Figure 4d,e and Figure 5d,e). Concentrations of $SO_2$ and $NO_2$ also do not show significant changes between the two seasons, with their decrease in the southern direction from Station 1 (Figure 4b,c and Figure 5b,c). The worst situation in a matter of TSP is in the northwestern part of the city near Stations 1 and 3 (Figures 4a and 5a).

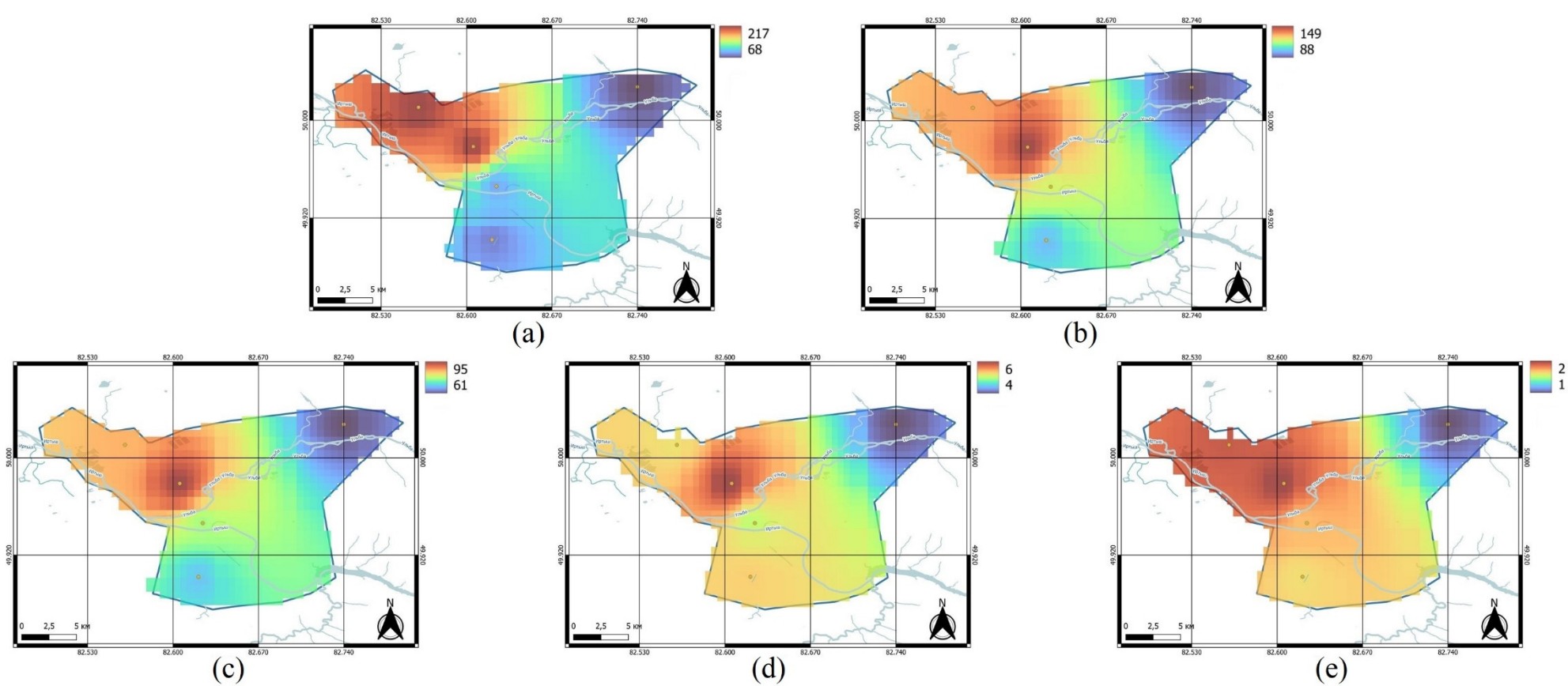

**Figure 4.** Spatial distribution patterns for (**a**) TSP, (**b**) SO$_2$, (**c**) NO$_2$, (**d**) HF, and (**e**) formaldehyde during the cold seasons ($\mu$g m$^{-3}$).

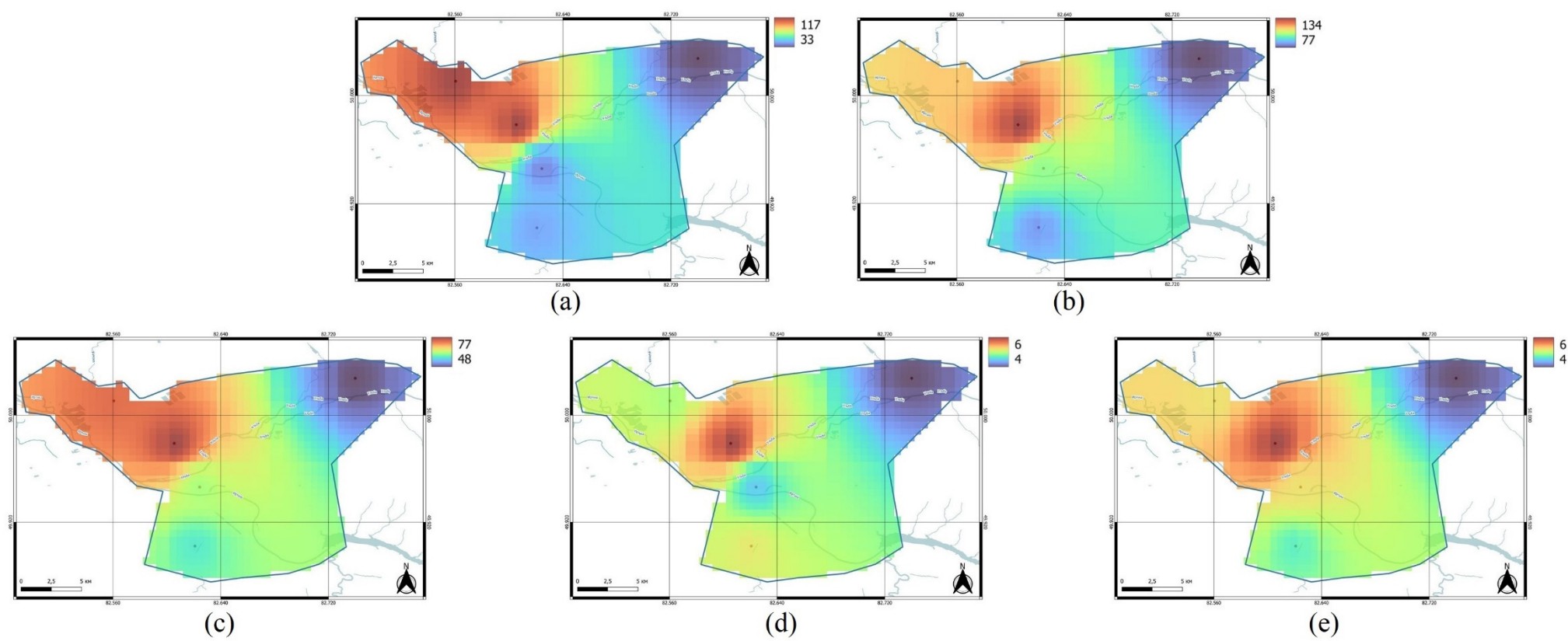

**Figure 5.** Spatial distribution patterns for (**a**) TSP, (**b**) SO$_2$, (**c**) NO$_2$, (**d**) HF, and (**e**) formaldehyde during the warm seasons (in μg m$^{-3}$).

**Table 2.** Descriptive statistics of air quality during the cold seasons (in μg m$^{-3}$).

| | WHO Limits (Daily Averaged) [51] | Kazakhstani Limits (Daily Averaged) [48] | | Station 1 | Station 2 | Station 3 | Station 4 | Station 5 |
|---|---|---|---|---|---|---|---|---|
| **TSP** | - | 150 | Mean (SD) | 217 (183) | 94 (92) | 218 (182) | 68 (66) | 81 (78) |
| | | | Median | 175 | 75 | 175 | 50 | 50 |
| | | | Range | 0–1350 | 0–550 | 0–1250 | 0–425 | 0–550 |
| **SO$_2$** | 40 | 50 | Mean (SD) | 150 (124) | 122 (108) | 134 (105) | 88 (53) | 101 (69) |
| | | | Median | 100 | 84 | 93 | 73 | 79 |
| | | | Range | 31–1641 | 14–1738 | 31–1318 | 0–410 | 0–519 |
| **CO** | 4000 | 3000 | Mean (SD) | 1548 (877) | 656 (686) | 1285 (1025) | 195 (371) | 376 (483) |
| | | | Median | 1500 | 500 | 1000 | 0 | 250 |
| | | | Range | 0–6250 | 0–3750 | 0–5750 | 0–2250 | 0–3500 |
| **NO$_2$** | 25 | 40 | Mean (SD) | 96 (54) | 77 (48) | 86 (50) | 61 (37) | 69 (45) |
| | | | Median | 85 | 68 | 75 | 53 | 58 |
| | | | Range | 0–380 | 5–315 | 0–362 | 5–323 | 0–365 |
| **Phenol** | - | 3 | Mean (SD) | 2.4 (1.8) | 1.8 (1.6) | 2.1 (1.7) | 1.2 (1.4) | 1.9 (1.7) |
| | | | Median | 2.0 | 1.5 | 1.8 | 0.8 | 1.5 |
| | | | Range | 0–12.8 | 0–11.0 | 0–20.5 | 0–20.5 | 0–16.0 |
| **HF** | - | 5 | Mean (SD) | 6.6 (3.5) | 5.6 (3.4) | 5.8 (3.4) | 4.6 (3.0) | 5.9 (3.8) |
| | | | Median | 6.3 | 5.0 | 5.3 | 4.0 | 5.3 |
| | | | Range | 0–25 | 0–25 | 0–25 | 0–18 | 0–29 |
| **HCl** | - | 100 | Mean (SD) | 49.2 (28.5) | 37.1 (28.5) | 45.5 (27.5) | 35.2 (28.6) | 34.9 (29) |
| | | | Median | 50.0 | 30.0 | 47.5 | 30.0 | 30.0 |
| | | | Range | 0–175.0 | 0–142.5 | 0–152.5 | 0–142.5 | 0–145 |
| **H$_2$SO$_4$** | - | 100 | Mean (SD) | 23.2 (25.4) | 12.5 (15.4) | 20.6 (22.1) | 7.6 (10.0) | 10.2 (12.7) |
| | | | Median | 15.0 | 7.5 | 15.0 | 5.0 | 7.5 |
| | | | Range | 0–280.0 | 0–157.5 | 0–220.0 | 0–165.0 | 0–172.5 |
| **Formald** | - | 10 | Mean (SD) | 2.6 (2.0) | 2.2 (1.8) | 2.5 (2.2) | 1.3 (1.6) | 2.1 (1.9) |
| | | | Median | 2.5 | 1.5 | 2.5 | 1.3 | 2.0 |
| | | | Range | 0–11.8 | 0–10.0 | 0–15.0 | 0–10.0 | 0–11.8 |
| **H$_2$S** | - | 0.8 | Mean (SD) | 2.0 (1.8) | 1.8 (2.1) | 1.4 (1.1) | 0.6 (0.7) | 1.0 (1.0) |
| | | | Median | 1.5 | 1.3 | 1.0 | 0.5 | 1.0 |
| | | | Range | 0–18.3 | 0–23.0 | 0–11.8 | 0–5.0 | 0–11.8 |

**Table 3.** Descriptive statistics of air quality during the warm seasons (μg m$^{-3}$).

| | WHO Limits (Daily Averaged) [51] | Kazakhstani Limits (Daily Averaged) [48] | | Station 1 | Station 2 | Station 3 | Station 4 | Station 5 |
|---|---|---|---|---|---|---|---|---|
| **TSP** | - | 150 | Mean (SD) | 118 (81) | 41 (42) | 117 (80) | 33 (41) | 47 (45) |
| | | | Median | 100 | 25 | 100 | 25 | 25 |
| | | | Range | 0–500 | 0–375 | 0–475 | 0–375 | 0–300 |
| **SO$_2$** | 40 | 50 | Mean (SD) | 135 (98) | 102 (63) | 116 (68) | 77 (30) | 85 (36) |
| | | | Median | 98 | 81 | 90 | 71 | 75 |
| | | | Range | 45–1100 | 13–490 | 43–468 | 13–256 | 0–353 |
| **CO** | 4000 | 3000 | Mean (SD) | 886 (607) | 149 (234) | 571 (517) | 19 (107) | 133 (219) |
| | | | Median | 750 | 0 | 500 | 0 | 0 |
| | | | Range | 0–5000 | 0–1750 | 0–2750 | 0–1250 | 0–1250 |
| **NO$_2$** | 25 | 40 | Mean (SD) | 78 (46) | 62 (38) | 74 (43) | 48 (29) | 58 (35) |
| | | | Median | 70 | 54 | 68 | 45 | 53 |
| | | | Range | 0–308 | 0–212 | 0–233 | 0–218 | 0–263 |
| **Phenol** | - | 3 | Mean (SD) | 2.2 (1.6) | 1.8 (1.6) | 2.0 (1.5) | 1.2 (1.1) | 2.0 (1.7) |
| | | | Median | 2.0 | 1.5 | 1.8 | 0.8 | 1.8 |
| | | | Range | 0–16.5 | 0–18.0 | 0–17.0 | 0–5.5 | 0–14.8 |
| **HF** | - | 5 | Mean (SD) | 6.2 (3.6) | 4.9 (3.1) | 5.3 (3.1) | 4.5 (3.0) | 5.5 (3.3) |
| | | | Median | 6.0 | 4.3 | 5.0 | 3.5 | 4.9 |
| | | | Range | 0–20.0 | 0–16.3 | 0–16.5 | 0–15.5 | 0–16.3 |
| **HCl** | - | 100 | Mean (SD) | 51 (34) | 39 (32) | 48 (32) | 42 (39) | 39 (34) |
| | | | Median | 48 | 33 | 45 | 30 | 30 |
| | | | Range | 0–167 | 0–175 | 0–148 | 0–173 | 0–145 |
| **H$_2$SO$_4$** | - | 100 | Mean (SD) | 14.6 (15.7) | 7.5 (6.4) | 12.1 (9.6) | 4.4 (4.4) | 6.9 (5.0) |
| | | | Median | 12.5 | 7.5000 | 10.0 | 2.5 | 5.0 |
| | | | Range | 0–222.5 | 0–52.5 | 0–102.5 | 0–30.0 | 0–33.0 |
| **Formald** | - | 10 | Mean (SD) | 6.8 (5.0) | 5.9 (4.7) | 5.9 (4.3) | 4.5 (3.4) | 5.3 (3.6) |
| | | | Median | 5.9 | 4.8 | 5.0 | 3.8 | 4.5 |
| | | | Range | 0–33.5 | 0–35.8 | 0–32.3 | 0–21.8 | 0–28.0 |
| **H$_2$S** | - | 0.8 | Mean (SD) | 1.5 (0.9) | 1.3 (0.8) | 1.2 (0.9) | 0.6 (0.6) | 0.9 (0.8) |
| | | | Median | 1.5 | 1.0 | 1.0 | 0.5 | 0.8 |
| | | | Range | 0–5.5 | 0–4.3 | 0–4.8 | 0–3.5 | 0–3.8 |

### 3.3. Principal Component Analysis

The correlation matrix was employed for each monitoring station for all 859 and 602 measurements (for cold and warm seasons, respectively) for determining relationships: in particular, pairs between contaminants and meteorological parameters. Figure 6 shows the averaged values for correlation coefficients among the monitoring stations. The analysis has not identified any correlation between the chemical and meteorological (including wind direction, wind speed, relative humidity, and temperature) parameters. Tables 4 and 5, representing the principal components (PCs), have been developed for contaminants measurements only to identify groups of the contaminants, which can be combined by their common origin or specific properties of their spatiotemporal distribution (Table 6). Bold values in Tables 4 and 5 denote high loadings of the contaminants to the calculated PCs. The eigenvalues of the identified PCs are all greater than 1.0, and according to the Kaiser criterion, these PCs have to be chosen [52]. The results did not show significant differences between the stations in general, which may indicate a relatively equal distribution of pollution within the city. Particular differences are described in the subsections for each PC below. It is important to note here that the correlation during the cold seasons is stronger than during the warm seasons (Figure 6). Two PCs have been identified for the cold season analysis for the monitoring Stations 1 and 2, while three PCs have been identified for other monitoring stations. Therefore, included parameters in PCs 1 and 2 for Stations 3–5 are almost the same as the parameters included in PC1 for Stations 1–2. PCA, for the warm season, has identified three PCs for Stations 2–5 and four PCs for Station 1.

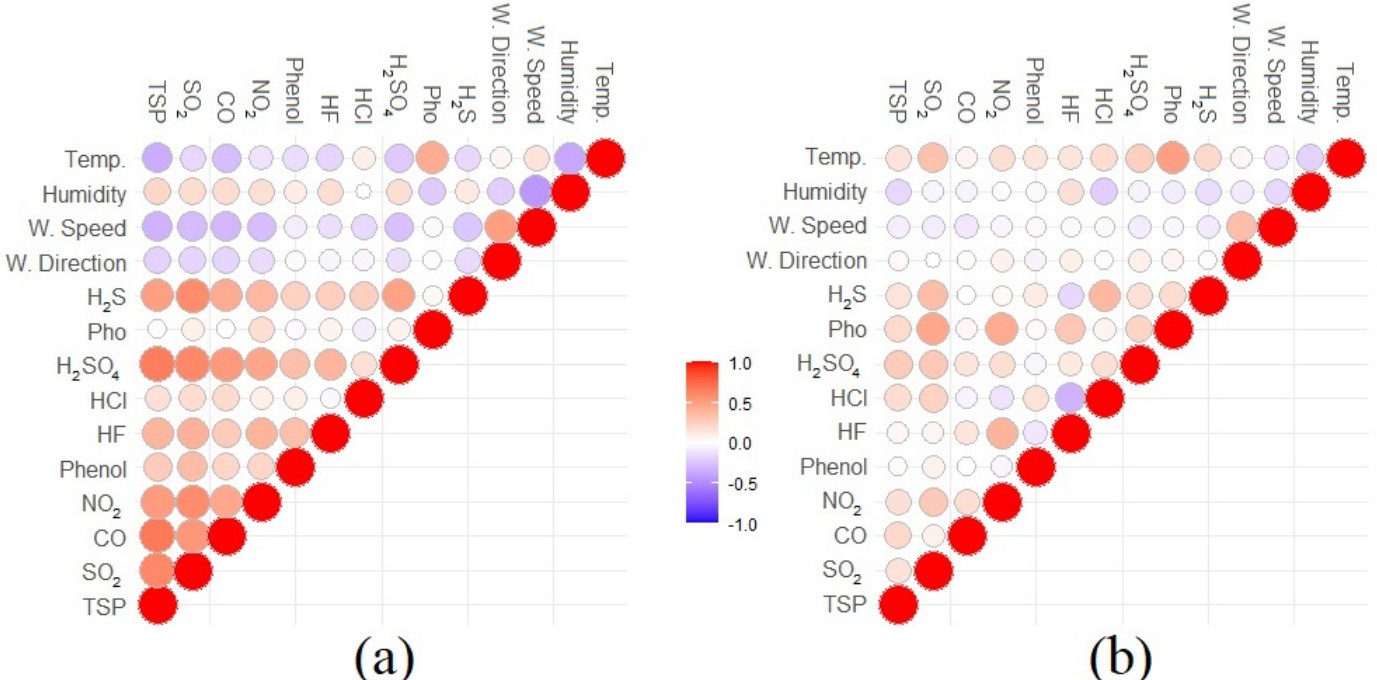

**Figure 6.** Pearson correlation matrix air quality parameters and climate characteristics for (**a**) the cold and (**b**) the warm seasons.

**Table 4.** Principal Component Analysis for the cold seasons.

| | Station 1 | | Station 2 | | Station 3 | | | Station 4 | | | Station 5 | | |
|---|---|---|---|---|---|---|---|---|---|---|---|---|---|
| | PC1 | PC2 | PC1 | PC2 | PC1 | PC2 | PC3 | PC1 | PC2 | PC3 | PC1 | PC2 | PC3 |
| TSP | **0.803** | 0.093 | **0.859** | −0.121 | **0.829** | 0.228 | 0.079 | **0.773** | 0.218 | −0.047 | **0.669** | 0.509 | 0.109 |
| $SO_2$ | **0.841** | −0.054 | **0.835** | −0.018 | **0.745** | 0.395 | 0.007 | **0.747** | 0.363 | 0.150 | **0.603** | 0.476 | 0.350 |
| CO | **0.633** | −0.023 | **0.796** | −0.267 | **0.800** | 0.237 | 0.065 | **0.718** | 0.068 | 0.015 | **0.681** | 0.387 | −0.001 |
| $NO_2$ | **0.748** | 0.198 | **0.772** | 0.197 | **0.580** | 0.485 | −0.216 | **0.523** | 0.260 | 0.465 | 0.406 | 0.315 | **0.573** |
| Phenol | **0.530** | 0.035 | 0.452 | −0.036 | 0.146 | **0.686** | 0.197 | 0.212 | **0.590** | −0.234 | 0.066 | **0.748** | −0.133 |
| HF | **0.560** | 0.443 | **0.545** | 0.340 | 0.232 | **0.776** | −0.078 | 0.304 | **0.671** | 0.172 | 0.107 | **0.763** | 0.159 |
| HCl | 0.248 | **−0.758** | 0.225 | **−0.650** | 0.361 | −0.155 | **0.684** | 0.493 | **−0.584** | −0.138 | **0.693** | −0.317 | −0.125 |
| $H_2SO_4$ | **0.835** | −0.060 | **0.809** | −0.046 | **0.804** | 0.197 | −0.045 | **0.695** | 0.348 | 0.009 | 0.467 | **0.580** | 0.180 |
| Formaldehyde | 0.101 | **0.558** | 0.092 | **0.747** | 0.175 | −0.199 | **−0.788** | −0.011 | −0.050 | **0.893** | −0.095 | −0.086 | **0.880** |
| $H_2S$ | **0.739** | −0.220 | **0.806** | −0.100 | **0.706** | −0.018 | 0.109 | **0.708** | −0.102 | 0.059 | **0.702** | 0.164 | 0.090 |
| Eigenvalue | 4.223 | 1.185 | 4.541 | 1.222 | 3.599 | 1.675 | 1.205 | 3.309 | 1.526 | 1.146 | 2.656 | 2.346 | 1.336 |
| % of variance | 42.231 | 11.847 | 45.409 | 12.223 | 35.986 | 16.751 | 12.049 | 33.089 | 15.264 | 11.457 | 26.559 | 23.465 | 13.363 |
| Cumulative % | 42.231 | 54.078 | 45.409 | 57.632 | 35.986 | 52.738 | 64.787 | 33.089 | 48.352 | 59.809 | 26.559 | 50.024 | 63.387 |

**Table 5.** Principal Component Analysis for the warm seasons.

| | Station 1 | | | | Station 2 | | | Station 3 | | | Station 4 | | | Station 5 | | |
|---|---|---|---|---|---|---|---|---|---|---|---|---|---|---|---|---|
| | PC1 | PC2 | PC3 | PC4 | PC1 | PC2 | PC3 | PC1 | PC2 | PC3 | PC1 | PC2 | PC3 | PC1 | PC2 | PC3 |
| TSP | 0.087 | 0.321 | **0.665** | −0.130 | 0.210 | 0.026 | **0.664** | 0.119 | −0.107 | **0.838** | 0.150 | 0.427 | **0.548** | 0.209 | 0.200 | **0.684** |
| $SO_2$ | **0.711** | 0.278 | −0.044 | −0.008 | **0.679** | 0.303 | 0.127 | 0.502 | **0.569** | 0.230 | **0.717** | 0.336 | 0.006 | **0.685** | 0.399 | 0.124 |
| CO | −0.113 | 0.023 | **0.804** | 0.143 | −0.195 | 0.032 | **0.725** | −0.519 | 0.339 | **0.521** | 0.185 | −0.014 | 0.393 | −0.066 | −0.206 | **0.776** |
| $NO_2$ | 0.187 | **0.751** | 0.303 | 0.094 | **0.727** | −0.335 | 0.107 | −0.276 | **0.744** | 0.158 | **0.679** | −0.200 | 0.063 | **0.699** | −0.280 | 0.059 |
| Phenol | 0.179 | −0.073 | 0.097 | **0.865** | −0.044 | 0.413 | −0.045 | 0.359 | −0.001 | −0.158 | 0.318 | 0.299 | **−0.666** | 0.005 | 0.376 | −0.165 |
| HF | −0.179 | **0.773** | 0.155 | −0.166 | 0.474 | **−0.637** | 0.057 | −0.522 | **0.588** | −0.114 | 0.521 | **−0.538** | 0.193 | 0.436 | **−0.618** | 0.007 |
| HCl | **0.645** | −0.484 | 0.054 | 0.054 | 0.099 | **0.757** | 0.220 | **0.691** | −0.087 | 0.273 | −0.045 | **0.806** | 0.121 | 0.038 | **0.779** | 0.134 |
| $H_2SO_4$ | 0.458 | −0.031 | 0.421 | −0.497 | 0.261 | 0.050 | **0.626** | 0.112 | 0.206 | **0.693** | 0.411 | 0.154 | 0.399 | 0.450 | 0.095 | 0.476 |
| Formaldehyde | 0.485 | **0.612** | −0.088 | 0.002 | **0.825** | −0.035 | 0.067 | 0.197 | **0.820** | 0.008 | **0.787** | 0.076 | 0.120 | **0.797** | 0.098 | 0.081 |
| $H_2S$ | **0.763** | 0.005 | 0.012 | 0.111 | 0.425 | **0.516** | 0.111 | **0.734** | 0.078 | 0.190 | 0.152 | **0.704** | −0.063 | 0.228 | **0.684** | 0.119 |
| Eigenvalue | 2.069 | 1.958 | 1.404 | 1.085 | 2.238 | 1.626 | 1.457 | 2.078 | 2.078 | 1.680 | 2.218 | 1.888 | 1.132 | 2.087 | 1.936 | 1.382 |
| % of variance | 20.693 | 19.578 | 14.037 | 10.850 | 22.377 | 16.260 | 14.567 | 20.779 | 20.778 | 16.797 | 22.177 | 18.880 | 11.325 | 20.872 | 19.360 | 13.817 |
| Cumulative % | 20.693 | 40.271 | 54.308 | 65.158 | 22.377 | 38.637 | 53.205 | 20.779 | 41.557 | 58.354 | 22.177 | 41.056 | 52.381 | 20.872 | 40.232 | 54.049 |

**Table 6.** Grouped PCs for cold and warm seasons.

| Cold Season | | Warm Season | |
|---|---|---|---|
| **Group 1** | **Group 2** | **Group 1** | **Group 2** |
| TSP | HCl | $SO_2$ | TSP |
| $SO_2$ | Formaldehyde | $NO_2$ | CO |
| CO | | Formaldehyde | Phenol |
| $NO_2$ | | HCl | $H_2SO_4$ |
| Phenol | | $H_2S$ | |
| HF | | HF | |
| $H_2SO_4$ | | | |
| $H_2S$ | | | |

### 3.3.1. PC1

PC1 of the cold season is characterised by high positive weight values for TSP, $SO_2$, CO, and $H_2S$ for every monitoring station. $NO_2$, phenol, HF, and $H_2SO_4$ may also be conditionally added to the list, as they have been formulated as PC 1 for Stations 1, 2, 3, and 4 and have been included in PC2 for Station 5. Altogether, the PC1 for Stations 1–2 and PC1 + PC2 for Stations 3–5 explain 47.7% of the total variance on average. As Figure 6 indicates, there is a strong positive correlation between TSP, $SO_2$, CO, and $H_2S$. $NO_2$ and $H_2SO_4$ also show a significant correlation with the listed contaminants. These ions are the major contributors to the total suspended particles. Additionally, these ions correlate with each other. It can be concluded that these contaminants have a shared source of their origin. The results of the appearance of TSP and CO in one group with $SO_2$ and $NO_2$ for the cold seasons and theirs separating for the warm seasons can be well explained by the fact that $SO_2$ and $NO_2$ are emitted mainly from combustion processes, while the sources for TSP are far more (for example, wind-driven and traffic-related re-suspension and biogenic sources) [53]. In addition, the metallurgy sector, which prevails in Ust-Kamenogorsk among emitters, can be characterised as the major source of $SO_2$ [54] and TSP (on behalf of $PM_{2.5}$) [55]. It is well known that the exposure of TSP-sulphur-derived contaminants in the air is believed to be representative of emissions from the combustion of fossil fuels, which increases the risk for bronchitis and some other respiratory disorders [56]. In addition, it is worth mentioning that nitrogen dioxide has been listed as an emerging pollutant causing morbidity and mortality [57].

PC1 of the warm season seems more uncertain, as there are no contaminants repeated in each of the monitoring stations. $SO_2$ (in four of five monitoring stations) and $NO_2$ and formaldehyde (in three of five) have been identified in PC1 as the most common contaminants. The contaminants $SO_2$, $NO_2$, and formaldehyde have been the same for the monitoring Stations 2, 4, and 5. This combination of the stations and the identified contaminants can be explained by the proximity of the monitoring stations to heat supply sources and highways [58] and the respective photochemical reactions in the atmosphere nearby [59]. In addition, the contaminants HCl and $H_2S$ have been identified as key contaminants in PC1 for Stations 1 and 3, which can be explained by emissions from the largest thermal power plant. A study [60] says that $H_2S$ is formed by the combustion of coal with high content of sulphur, which is an exact characteristic of Kazakhstani coal. The loading of HCl within this PC can be explained by the emissions from Kazzinc LLP according to the technological scheme of the enterprise.

### 3.3.2. PC2

PC2 in warm seasons compensates for the above-mentioned contaminants in the stations which previously have not been listed as belonging to PC1. For example, for Stations 2, 4, and 5, the list of contaminants includes HCl and $H_2S$. As mentioned above, the source of HCl emissions can be the metallurgy industry, which uses acid for production purposes. H2S can be emitted due to coal combustion. Station 2 is located in the central part of the city at a considerable distance from industrial facilities. However, sources of

emissions located at a reasonable elevation can disperse HCl and $H_2S$ to the centre of the city. For Stations 4 and 5, the presence of the parameters in this PC can be explained by the proximity of the northeastern industrial zone and the Sogrinskaya and the left-bank thermal power plants.

For Stations 1 and 3, the list of contaminants has been fulfilled by $SO_2$, $NO_2$, and formaldehyde. The northern industrial zone is a dominant emitter for Station 3, which can easily explain the presence of $SO_2$ and $NO_2$ in this PC. HF has been identified as the key contaminant for this PC in four of five monitoring stations. Figures 4 and 5 show that HF is mainly concentrated in the area of the northern industrial zone and then is evenly distributed throughout the city during both the cold and warm seasons. The thermal power plants do not monitor the concentrations of HF in their emissions, while there are a number of research [61,62] and official reports [63,64] showing the coal-fired stations as the largest anthropogenic sources of HF. The situation for formaldehyde looks the same as for HF. No enterprises monitor this chemical, while the direction of spatial distribution shows a similar direction.

### 3.3.3. PC3

PC3, for the warm season, has grouped TPS and CO, which indicates their shared source of origin, even though there is no household heating during the comfortable weather conditions and emissions from the central heating are expected to be minimal. Moreover, it is expected that nature might neglect the effect of air pollution by these contaminants, which seems to be a reason for such difference in the rate of these contaminants in the cold (the most impactful contaminants) and warm (the least impactful contaminants) seasons. In addition, phenol (in Stations 1 and 4) and $H_2SO_4$ (in Stations 2 and 3) have been revealed as belonging to this PC. Sulphuric acid is actively used in metallurgy, and this parameter shows the shared source in this PC with TPS and CO. The appearance of phenol in this PC requires additional studies, as its spread is unusual for metallurgy. This PC can be explained by a geographical location, as it focuses on the air conditions in the central part of the city. While emissions can be dispersed from the northern industrial zone according to Figures 4 and 5, the presence of phenol can be explained by intensive traffic [65].

Table 6 summarises a conditional grouping of identified PCs, results of the spatiotemporal assessment, hierarchical clustering analysis, and respective chemicals. Group 1 for the cold seasons can be explained by the intensive use of the fossil fuel fired by all users: power plants, industry, and households coupled with meteorological conditions (based on Figure 6). This mix ensures a variety of pollutants are released into the atmosphere by the burning of different types of fuel. The same group for the warm seasons indicates and highlights the problem of coal consumption, mainly by industrial enterprises, which is indicated by the uniform seasonal distribution of $NO_2$ and $SO_2$ [4]. Group 2 for the cold seasons can be explained by specific industrial processes and the release of the associated contaminants into the atmosphere. Group 2 for the warm seasons could indicate a contribution of traffic to air pollution. While local authorities have attempted to explain air quality issues by a high density of motor transport, the results of this study confirm outcomes of previous research on heavy air pollution in the city caused by non-transport-related sources [66]: that this factor has a low significance comparatively with the impact of industrial activities, especially burnt-coal based.

### 4. Conclusions

Ust-Kamenogorsk is one of the most important industrial centres of Kazakhstan, where non-ferrous metals are produced and exported abroad for the largest world companies. The active consumption of coal and raw material orientation of the industrial enterprises makes Ust-Kamenogorsk one of the most polluted industrial cities in the world. This study, for the first time, aimed to analyse spatiotemporal patterns of air pollution in the city and to identify potential sources of apportionment using multivariate statistical techniques. The results show that the combination of large enterprises and coal-fired thermal power plants

severely affects air quality in Ust-Kamenogorsk. The average concentrations of $SO_2$ and $NO_2$ for the entire study period exceeded the standards of WHO and Kazakhstan within the whole city all year round. The major emitters are located in the northern industrial zone with two huge metallurgy factories and one thermal power plant. The Principal Component Analysis revealed that emissions of TSP, $SO_2$, CO, $H_2S$, $NO_2$, HF, and $H_2SO_4$ with high likelihood are of industrial origin, which is significantly aggravated during the cold seasons. While official reports show a decrease in industrial emissions, air quality has not been improved even during the warm seasons when households stop heating with the expected elimination of air pollution.

The results of this study can be valuable for decision-makers in developing and applying respective strategies and actions for a decrease in air pollution and elimination of the social and environmental effects. The main limitation of the research is the difficulty in determining the ratio of the contributions of the major emitters to the high level of air pollution. Future research will focus on a detailed investigation of the composition of unstudied contaminants (particularly, particulate matters) in both ambient air and the zones of industrial enterprises. This research should be combined with the modelling of source profiles or fingerprints.

The air quality issues in Ust-Kamenogorsk most probably have been caused by a combination of the factors: weak environmental regulation and control, the influence of large companies on the legislative process through large professional associations, and an outdated and energy-intensive industry. The authors hope that the outcomes of this and other studies would be a sufficient research basis for authorities to develop the right strategy for air management in the region.

**Author Contributions:** Conceptualisation, D.A. and I.R.; methodology, I.R.; software, I.R.; validation, O.P., S.G. and G.M.; formal analysis, I.R. and D.A.; investigation, I.R., O.P., S.G. and G.M.; resources, D.A. and V.Z.; data curation, D.A. and V.Z.; writing—original draft preparation, I.R., D.A. and V.Z.; writing—review and editing J.J.K.; visualisation, I.R.; project administration, D.A.; funding acquisition, D.A. All authors have read and agreed to the published version of the manuscript.

**Funding:** This work has been funded by a grant from the Ministry of Education and Science of the Republic of Kazakhstan No. AP08053440 Research of emissions major sources of harmful substances into atmospheric air of Ust-Kamenogorsk with the cause identification of high hydrogen sulphide content.

**Institutional Review Board Statement:** Not applicable.

**Informed Consent Statement:** Not applicable.

**Data Availability Statement:** Data have been provided by The National Hydrometeorological Service of Kazakhstan "Kazhydromet" upon a special request.

**Acknowledgments:** The authors are thankful to Kazhydromet for providing the data on pollutants concentrations.

**Conflicts of Interest:** The authors declare no conflict of interest.

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
