# Peer review of "Spatiotemporal Patterns of Air Pollution in an Industrialised City—A Case Study of Ust-Kamenogorsk, Kazakhstan"

_atmosphere, doi:10.3390/atmos13121956_

Round 1

Reviewer 1 Report

Comments to the Author

It was my pleasure to review the manuscript entitled Spatiotemporal patterns of air pollution in an industrialized city – a case study of Ust-Kamenogorsk, Kazakhstan

Assanov, D., et al.,

In this study the authors used ten monitored pollutants, which contain TSP, SO2, CO, H2S, NO2, HF, HCl, H2SO4, phenol, and Formaldehyde from five different monitor stations in Ust-Kamenogorsk City in Kazakhstan for the period 2017 to 2021.  Hierarchical cluster analysis was used to assess the spatiotemporal patterns of the air quality of the city and PCA were used to discuss the possible sources of these pollutants for different stations. But there are lots of structures or texts should be modified. 

Major comments:

1.       In the abstract, from line 15 to 18, which contaminants were discussed? They should be give here.

2.       In the introduction, it is better to good summary of the relative researches about the air quality in the study area or concerns the same contaminants worldwide to illustrate the meaningful of this study.   

3.       For Figure 1, the right parts do not clearly displayed. Modify it or another better way to illustrate the sampling station and the surround environments.

4.       Line 117 to 122: It is better to state the sampling locations from S1 to S5 in the figure 1 caption.

5.       For Figure 3, the Kazakhstani limits (blue line) were not displayed in the figure. Please modify them.

6.       For the part of “3.3 principal component analysis”, line 226 to231. Were the daily pollutants values used to do the correlation between the pollutants and meteorological parameters? How about the different stations? Did it have obvious differences or not?

7.       Figure 4a is for cold season and figure 4b id for warm season? But the caption of figure 4a and 4b both showed the cold seasons. Please modify it.  

8.       Also for the “3.3 principal component analysis” the PCA results should be described for different stations by different sources not by PC1 PC2 PC3 like that. The texts should be thoroughly changed.  

9.       Please have this manuscript to be edited by a native English professor. The edition should include the whole text and grammar.

Author Response

Response to Reviewer 1 Comments

The authors thank the reviewer for the time spent to comment and improve the manuscript. We have revised the manuscript accordingly, taking the comments from the reviewer into consideration. Below are answers to the comments and description of actions taken: The black text is comments from the reviewer, and the Red text is our response.

Point 1: In the abstract, from line 15 to 18, which contaminants were discussed? They should be give here.

Thank you for your comment. The text has been modified in lines 15-31: “Air quality issues still affect the quality of life for people in industrialised cities around the world. The investigations should include the identification of the sources of the pollution and its distribution in space and time. This work is the first attempt to perform identification of the sources of pollution in Ust-Kamenogorsk city in Kazakhstan. Analysis of retrospective data (including ten variables (TSP, SO2, CO, NO2, phenol, HF, HCl, H2SO4, formaldehyde, H2S) from five monitoring stations for the period 2017-2021) using multivariate statistical methods and hierarchical cluster analysis has been performed to assess spatiotemporal patterns of air quality of the city. The results indicate that the contamination patterns can be grouped into two categories: cold and warm seasons. The study revealed the dangerous concentrations of NO2 and SO2 exceeded the limits by 2-3 and 1.5-2 times, independently of the seasonality. Averaged concentrations of TSP slightly exceeded the established limits for the most industrialised part of the city. Concentrations of HF and formaldehyde significantly rose during the cold seasons compared to the warm seasons. Other chemical parameters significantly depend on the seasonality and locations of the sampling points. The major reason for air pollution is twofold – the use of a burnt-coal throughout the year for electricity and heat generation (especially during the cold seasons) and the high density of the heavy metallurgy industry in the city. The principal component analysis confirms a high loading of industrial sources of air pollution on both spatial and seasonal dimensions.”.

Point 2: In the introduction, it is better to good summary of the relative researches about the air quality in the study area or concerns the same contaminants worldwide to illustrate the meaningful of this study.

Thank you for this suggestion. The Introduction section has been revised accordingly and now looks as follows:

(lines 64-78) “One of the problematic cities in the country, Ust-Kamenogorsk, is located in a major mining and smelting area and has been rated among the worst cities regarding the problems of ecology according to World Bank [14]. Extreme pollution events from accidental emissions are not rare in the city [15]. Descriptive studies of poor air quality in the city have said that the city has been rated the most SO2-polluted city in Kazakhstan and among the five most-polluted NO2 and O3 cities in Kazakhstan from 2011 to 2017 [12]. A study about the impact of COVID-19 lockdown on air quality in the city has revealed that the level of CO has decreased by 21–23 % with the increase of the TSP level by 13–21 %, and had no significant effect on SO2 and NO2 concentrations in the city [16]. Elevated levels of trace elements, particularly Ba, Mn, Pb, V, and Zn, in the blood of city residents of Ust-Kamenogorsk have been found [17], which could indicate a severe impact on the industrial activities of the city. The known studies are limited by their descriptive origin, while there is a demand for deep investigations of the relationships between a high level of air pollution, weak environmental regulation, pyrometallurgical processes and dependence on coal-burnt energy.”…

(lines 92-130): “According to an opinion from [24], preliminary investigations play a core role in the identification of the sources of pollution to drive “the wheel of progress: Sources ⇒ Effects ⇒ Regulation ⇒ Control”. There are known different ways to investigate potential sources apportionment: emission inventories [25], inverse modelling [26], artificial neural networks [27], receptor modelling methods [28], air quality models [25], a combination of the different approaches and models [29], application of positive matrix factorisation [30], and even chemical analysis of biomonitoring species [31]. However, all of the listed methods demand a very detailed dataset with a number of monitored parameters. This limitation is a serious obstacle in the conditions of many developing countries, including Kazakhstan, where the system of permanent monitoring with extended parameters, such as PMs, has still been established [32].

Under these conditions, one of the efficient ways to investigate sources apportionment with the following understanding of the needed steps for air quality management can be multivariate statistical techniques, particularly the Principal Component Analysis (PCA) and the Cluster Analysis. Also, using analysis of scores and loadings in PCA can give a representation of potential chemical reactions in ambient air [33]. The approach to applying PCA for the assessment of patterns in air quality has already been applied several times. For instance, Dominick et al. [34] have aimed to investigate possible sources of air pollutants and spatial patterns in Malaysia using the same techniques as the authors plan to do with grouping particulate contaminants in principal components. Application of the PCA to find the correlation between gaseous pollutant concentrations, meteorological factors and potential sources of pollution has identified the contribution of combustion- and non-combustion-related emitters in Greece [35] or in India [36]. The tool has been used for the same purpose also by Azid et al. in the context of the prediction of air pollution [37]. PCA has supported the assessment and identification of the sources of air pollution in India, where complex industrial activities exist [38]. Revealing information about the sources and mechanisms of air pollution in Madrid and visualising their spatial distribution using PCA and the geostatistical method has been carried out by Núñez-Alonso et al. [39]. Particular attention has been paid to the investigation of the presence of metals only in particulate matters coupled with multivariate statistical techniques in an Iranian industrial city [40]. For example, a combination of the analysis of the chemical composition of fine particulate matters with the focus on metals presence with Absolute Principal Component Analysis let to explain identified pollutants to their sources in the USA [41].

The aim of this study is to analyse key factors impacting air quality in Ust-Kamenogorsk using the available dataset for the period 2017-2021 by multivariate statistical techniques on spatial and temporal scales. This approach lets to investigate potential sources of apportionment using the large but limited, a dataset of the observations in the city for the first time.”

Point 3: For Figure 1, the right parts do not clearly displayed. Modify it or another better way to illustrate the sampling station and the surround environments.

Thank you for this comment. Modified.

Point 4: Line 117 to 122: It is better to state the sampling locations from S1 to S5 in the figure 1 caption.

Thank you for this comment. We have used both ways: we have captioned names of the observation stations in the Figure 1 and have also explained their locations in the text to show their location regarding the major emitters.

Point 5: For Figure 3, the Kazakhstani limits (blue line) were not displayed in the figure. Please modify them.

Thank you for this comment Now the color of the line has been changed to increase its visibility.

Point 6: For the part of “3.3 principal component analysis”, line 226 to231. Were the daily pollutants values used to do the correlation between the pollutants and meteorological parameters? How about the different stations? Did it have obvious differences or not.

Thank you for this question. Yes, the daily-averaged values were used for the PCA performance. You can find this inforemation in the sub-section 2.3. Data management and methodological framework: (lines 223-227) “The fourth step of this study was to perform PCA based on the correlation matrix. The correlation matrix was built using daily averaged values of the studied contaminants coupled with the meteorological parameters, while the PCA was done using chemical parameters only on a daily averaged measurements basis.”

The results of the PCA for each of the stations with the following description of special characteristics of PCs can be found in each sub-section of the 3.3. Principal component analysis. We have revised the text, added respective appendicies, and added some clarifications in the text as follows (lines 296-311):

“Table 3, representing the principal components (PCs), has been developed for contaminants measurements only to identify groups of the contaminants, which can be combined by their common origin or specific properties of their spatiotemporal distribution (Table 4). The eigenvalues of the identified PCs are all greater than 1.0, and according to the Kaiser criterion, these PCs have to be chosen [53]. The results of correlation matrices and the results of PCA for each station can be found in Appendices B and C. The results did not show significant differences between the stations in general, which may indicate a relatively equal distribution of pollution within the city. Particular differences are described in the sub-sections for each PC below. It is important to note here that the correlation during the cold seasons is stronger than during the warm seasons (Figure 5). Two PCs have been identified for the cold season analysis for the monitoring Stations 1 and 2, while three PCs have been identified for other monitoring stations. Therefore, included parameters in PCs 1 and 2 for Stations 3-5 are almost the same as the parameters included in PC1 for Stations 1-2. PCA, for the warm season, has identified three PCs for Stations 2-5 and four PCs for Station 1.”.

Point 7: Figure 4a is for cold season and figure 4b id for warm season? But the caption of figure 4a and 4b both showed the cold seasons. Please modify it.

Thank you for your careful reading. Fixed.

Point 8: Also for the “3.3 principal component analysis” the PCA results should be described for different stations by different sources not by PC1 PC2 PC3 like that. The texts should be thoroughly changed.

Thank you for this comment. As this section is named “3. Results and Discussion”, we consider that each sub-section presents the results with the following interpretation of the results of the particular analysis, in this case - the PCA. Due to this reason we can not agree with your point. However, we can see from the results of the PCA that there are the shared sources of the pollution, which have already been indicated in the description of the PCs. We have revised the whos section with a more detailed description of the obtained results with the concluding paragraph of the section (lines 390-404: “Table 4 summarises a conditional grouping of identified PCs, results of the spatiotemporal assessment, hierarchical clustering analysis, and respective chemicals. Group 1 for the cold seasons can be explained by the intensive use of the fossil fuel-fired by all users: power plants, industry, and households coupled with meteorological conditions (based on Figure 5). This mix ensures a variety of pollutants are released into the atmosphere by the burning of different types of fuel. The same group for the warm seasons indicates and highlights the problem of coal consumption, mainly by industrial enterprises, which is indicated by the uniform seasonal distribution of NO2 and SO2 [4]. Group 2 for the cold seasons can be explained by specific industrial processes and the release of the associated contaminants into the atmosphere. Group 2 for the warm seasons could indicate a contribution of traffic to air pollution. While local authorities have attempted to explain air quality issues by a high density of motor transport, the results of this study confirm outcomes of previous research that heavy air pollution in the city caused by non-transport-related sources [67] that this factor has a low significance comparatively with the impact of industrial activities, especially burnt-coal based.”.

Point 9: Please have this manuscript to be edited by a native English professor. The edition should include the whole text and grammar.

Thank you for your suggestion. The text has been revised and edited.

Reviewer 2 Report

The authors used multivariate statistical techniques to analyse retrospective data (2017-2021) and form a hierarchical cluster analysis to assess the air pollution situation and spatial and temporal patterns of urban air quality in the city of Ust-Kamenogorsk, Kazakhstan.The paper is well argued, but the following comments should be considered for a better reading:

1. The contribution and novelty of the paper should be described in the introduction.

2.The application of spatio-temporal models of air pollution has a long history. The authors' review of current research on air pollutant concentration pollution in the introductory section is not comprehensive, for example, citing research on PM2.5 concentration prediction.See for example “PM2.5 volatility prediction by XGBoost-MLP based on GARCH models”and “Prediction of Air Pollutant Concentration Based on One-Dimensional Multi-Scale CNN-LSTM Considering Spatial-Temporal Characteristics”.

3.The authors should have added a description of the results in sections 3.1 and 3.2, e.g. some detailed explanation of Figure 2.

4.Table 2 should be a three-line table.Rows 242, 288 and 310 should be in a single heading as in 3.3.1 PC1.

5.Tables 3a and b are proposed to be represented by correlation charts.

6.Explain the reasons for the difference in factors between groups I and II in the warm and cool seasons.

7.A discussion section should be added to compare current and existing research.

8.Row 201 has an extra \.

Author Response

Response to Reviewer 2 Comments

The authors thank the reviewer for the time spent to comment and improve the manuscript. We have revised the manuscript accordingly, taking the comments from the reviewer into consideration. Below are answers to the comments and description of actions taken: The black text is comments from the reviewer, and the Red text is our response.

Point 1: The contribution and novelty of the paper should be described in the introduction.

Thank you very much for this note. We have identified the strenghtnes of our paper comparatively with already done publication on air quality issues in the region and clarified the in the text respectively: (lines 64-78) “ One of the problematic cities in the country, Ust-Kamenogorsk, is located in a major mining and smelting area and has been rated among the worst cities regarding the problems of ecology according to World Bank [14]. Extreme pollution events from accidental emissions are not rare in the city [15]. Descriptive studies of poor air quality in the city have said that the city has been rated the most SO2-polluted city in Kazakhstan and among the five most-polluted NO2 and O3 cities in Kazakhstan from 2011 to 2017 [12]. A study about the impact of COVID-19 lockdown on air quality in the city has revealed that the level of CO has decreased by 21–23 % with the increase of the TSP level by 13–21 %, and had no significant effect on SO2 and NO2 concentrations in the city [16]. Elevated levels of trace elements, particularly Ba, Mn, Pb, V, and Zn, in the blood of city residents of Ust-Kamenogorsk have been found [17], which could indicate a severe impact on the industrial activities of the city. The known studies are limited by their descriptive origin, while there is a demand for deep investigations of the relationships between a high level of air pollution, weak environmental regulation, pyrometallurgical processes and dependence on coal-burnt energy.”

Lines (92-102): “According to an opinion from [24], preliminary investigations play a core role in the identification of the sources of pollution to drive “the wheel of progress: Sources ⇒ Effects ⇒ Regulation ⇒ Control”. There are known different ways to investigate potential sources apportionment: emission inventories [25], inverse modelling [26], artificial neural networks [27], receptor modelling methods [28], air quality models [25], a combination of the different approaches and models [29], application of positive matrix factorisation [30], and even chemical analysis of biomonitoring species [31]. However, all of the listed methods demand a very detailed dataset with a number of monitored parameters. This limitation is a serious obstacle in the conditions of many developing countries, including Kazakhstan, where the system of permanent monitoring with extended parameters, such as PMs, has still been established [32].”

Lines (126-130): “The aim of this study is to analyse key factors impacting air quality in Ust-Kamenogorsk using the available dataset for the period 2017-2021 by multivariate statistical techniques on spatial and temporal scales. This approach lets to investigate potential sources of apportionment using the large but limited, a dataset of the observations in the city for the first time.”.

Point 2: The application of spatio-temporal models of air pollution has a long history. The authors' review of current research on air pollutant concentration pollution in the introductory section is not comprehensive, for example, citing research on PM2.5 concentration prediction. See for example “PM2.5 volatility prediction by XGBoost-MLP based on GARCH models” and “Prediction of Air Pollutant Concentration Based on One-Dimensional Multi-Scale CNN-LSTM Considering Spatial-Temporal Characteristics”.

Thank you very much for this suggestion. Other reviewers have also pointed out the need to provide better historical background on the attempts to identify the sources of the pollution by different methods. The text has been revised and expanded significantly: (lines 92-125) “According to an opinion from [24], preliminary investigations play a core role in the identification of the sources of pollution to drive “the wheel of progress: Sources ⇒ Effects ⇒ Regulation ⇒ Control”. There are known different ways to investigate potential sources apportionment: emission inventories [25], inverse modelling [26], artificial neural networks [27], receptor modelling methods [28], air quality models [25], a combination of the different approaches and models [29], application of positive matrix factorisation [30], and even chemical analysis of biomonitoring species [31]. However, all of the listed methods demand a very detailed dataset with a number of monitored parameters. This limitation is a serious obstacle in the conditions of many developing countries, including Kazakhstan, where the system of permanent monitoring with extended parameters, such as PMs, has still been established [32].

Under these conditions, one of the efficient ways to investigate sources apportionment with the following understanding of the needed steps for air quality management can be multivariate statistical techniques, particularly the Principal Component Analysis (PCA) and the Cluster Analysis. Also, using analysis of scores and loadings in PCA can give a representation of potential chemical reactions in ambient air [33]. The approach to applying PCA for the assessment of patterns in air quality has already been applied several times. For instance, Dominick et al. [34] have aimed to investigate possible sources of air pollutants and spatial patterns in Malaysia using the same techniques as the authors plan to do with grouping particulate contaminants in principal components. Application of the PCA to find the correlation between gaseous pollutant concentrations, meteorological factors and potential sources of pollution has identified the contribution of combustion- and non-combustion-related emitters in Greece [35] or in India [36]. The tool has been used for the same purpose also by Azid et al. in the context of the prediction of air pollution [37]. PCA has supported the assessment and identification of the sources of air pollution in India, where complex industrial activities exist [38]. Revealing information about the sources and mechanisms of air pollution in Madrid and visualising their spatial distribution using PCA and the geostatistical method has been carried out by Núñez-Alonso et al. [39]. Particular attention has been paid to the investigation of the presence of metals only in particulate matters coupled with multivariate statistical techniques in an Iranian industrial city [40]. For example, a combination of the analysis of the chemical composition of fine particulate matters with the focus on metals presence with Absolute Principal Component Analysis let to explain identified pollutants to their sources in the USA [41].”

Point 3: The authors should have added a description of the results in sections 3.1 and 3.2, e.g. some detailed explanation of Figure 2.

Thank you for this suggestion. The section has been extended and now looks as follow: (lines 242-280) “Table 2 presents the results of measurements of air quality from the monitoring stations in the city according to the limits established by the Kazakhstani government and recommended by WHO. It is clearly seen that heavy industry and coal-burnt energy cause the permanent exceeding of the permissible daily values along the city during both cold and warm seasons. The worst situation remains for NO2 and SO2, which are significantly higher than both limits (by 2-3 times for NO2 and by 1.5-2 times for SO2) during the whole period of analysis (Figure 3). Averaged concentrations of TSP slightly exceeded the established limits for Stations 1 and 3 during the cold seasons, with the peak values during January months in 2017 and 2018 (Figure 3), while during the warm seasons, they can be characterised as safe. Concentrations of HF and H2S show slight excess over the recommended concentrations along all stations (except Station 4) during both seasons. However, the presence of HF dropped below the limit line after January 2020 (Figure 3). Surprisingly, the averaged and median concentrations of CO have not shown exceeding values, except several daily exceedings of the parameter were recognised during the cold seasons. In general, the concentrations of the pollutants show a slightly descending trend while they still remain much above the permissible level. It can be explained by the report from the National Statistics Committee, which claims that industrial emissions decreased from 29 to 27.9 kt/y for SO2 and from 10.8 to 10.4 kt/y for CO for the period between 2017 and 2021 [47].

It is fair to note that the recently updated Kazakhstani limits [48] have not followed recommendations from international standards [49,50] and experts [51]. The limits for main pollutants have not been revised and still are above the recommendations from WHO [52].

The spatial distribution of the contaminants is presented in Figure 4. There are plotted only TSP, SO2, NO2, HF, and formaldehyde show the most significant patterns in their dispersion within the city. It is clearly seen that the major emitter of the city is located near Station 1 and represents the northern industrial zone with two huge metallurgy factories and one thermal power plant. Surprisingly, the safest location in the city regarding concentrations of the pollutants is near Station 4, despite its close location to the northeastern industrial zone. The main difference between seasons is the presence of pollution near Station 5, in the southern direction from the major emitters to the downtown, located on the left-bank part of the city with its own thermal power plant. While concentrations of HF and formaldehyde look high, comparatively with the centre of the emissions during the cold seasons, the presence of these contaminants during the warm seasons looks safe and strives for minimal values within the city (Figure 4d-e). Concentrations of SO2 and NO2 also do not show significant changes between the two seasons, with their decrease in the southern direction from Station 1 (Figure 4b-c). The worst situation in a matter of TSP is in the north-western part of the city near Stations 1 and 3 (Figure 4a).”.

Point 4: Table 2 should be a three-line table.Rows 242, 288 and 310 should be in a single heading as in 3.3.1 PC1.

Thank you very much for this correction. The abovementioned issues have been fixed.

Point 5: Tables 3a and b are proposed to be represented by correlation charts.

Thank you for this suggestion. This point has been considered and the tables have been modified into the correlations charts

Point 6: Explain the reasons for the difference in factors between groups I and II in the warm and cool seasons.

Thank you for this note. The text has been updated as follows: (lines 390-404) “Table 4 summarises a conditional grouping of identified PCs, results of the spatiotemporal assessment, hierarchical clustering analysis, and respective chemicals. Group 1 for the cold seasons can be explained by the intensive use of the fossil fuel-fired by all users: power plants, industry, and households coupled with meteorological conditions (based on Figure 5). This mix ensures a variety of pollutants are released into the atmosphere by the burning of different types of fuel. The same group for the warm seasons indicates and highlights the problem of coal consumption, mainly by industrial enterprises, which is indicated by the uniform seasonal distribution of NO2 and SO2 [4]. Group 2 for the cold seasons can be explained by specific industrial processes and the release of the associated contaminants into the atmosphere. Group 2 for the warm seasons could indicate a contribution of traffic to air pollution. While local authorities have attempted to explain air quality issues by a high density of motor transport, the results of this study confirm outcomes of previous research that heavy air pollution in the city caused by non-transport-related sources [67] that this factor has a low significance comparatively with the impact of industrial activities, especially burnt-coal based.”.

Point 7: A discussion section should be added to compare current and existing research.

Thank you for your suggestion. The comparison has been partly covered in the Introduction and in the paragraph above. Some implications have also been made in the Conclusion: (lines 403-426) “Ust-Kamenogorsk is one of the most important industrial centres of Kazakhstan, where non-ferrous metals are produced are exported abroad for the largest world companies. Active consumption of coal and raw material orientation of the industrial enterprises make Ust-Kamenogorsk one of the most polluted industrial cities in the world. This study, for the first time, aimed to analyse spatiotemporal patterns of air pollution in the city and to identify potential sources of apportionment using multivariate statistical techniques. The results show that the combination of large enterprises and coal-fired thermal power plants severely affects air quality in Ust-Kamenogorsk. The average concentrations of SO2 and NO2 for the entire study period exceeded the standards of WHO and Kazakhstan within the whole city all year round. The major emitters are located in the northern industrial zone with two huge metallurgy factories and one thermal power plant. The principal component analysis revealed that emissions of TSP, SO2, CO, H2S, NO2, HF, and H2SO4 with high likelihood are of industrial origin, which is significantly aggravated during the cold seasons. While official reports show a decrease in industrial emissions, air quality has not been improved even during the warm seasons when households stop heating with the expected elimination of air pollution.

The results of this study can be valuable for decision-makers in developing and applying respective strategies and actions for a decrease in air pollution and elimination of the social and environmental effects. The main limitation of the research is the difficulty in determining the ratio of the contributions of the major emitters to the high level of air pollution. Future research will focus on a detailed investigation of the composition of unstudied contaminants (particularly, particulate matters) in both ambient air and the zones of industrial enterprises. This research should be combined with the modelling of source profiles or fingerprints.”.

Point 8: Row 201 has an extra \.

Thank you for your point. The text has been modified now and looks as follow: “Table 2 presents the results of measurements of air quality from the monitoring stations in the city according to the limits established by Kazakhstani government and recommended by WHO.”.

Reviewer 3 Report

Comments to the authors:

In this study, the authors studied the spatiotemporal pattern utilizing different statistical analyses to evaluate the air pollution conditions in Ust-Kamenogorsk, Kazakhstan, an industrialized city. Overall, the manuscript has a concrete structure, with promising results and discussions. However, there are a few comments to be addressed.

Abstract: Too general and I cannot identify the research gap that you are addressing. Please revise. 

Introduction: Major revision is required. The research gap and the research objectives were not clear in the submission. Line 92-93: Not professionally written. Please provide more information. Provide a more detailed literature review about the conditions and highlight what is your contributions.

Study area: Fig.1 better naming should be provided. Also, provide the scale bar on the map. 

Results and Discussions: Fig.2: Very hard to read. Suggest making two rows for better illustration. Fig 3: Similar issues: Poor arrangement. Please find a better arrangement method.  Perhaps Table 2 can be better illustrated using box plots.

Good discussion in section 3.3

Table 4: Consider coloring the bolded value.

Fig 4: Again, poor arrangement and low resolution of the figures.

There should be a separate section for discussion of findings. Further, findings must be compared with previous researchers’ work. Unique contributions of research should be highlighted.

Grammatical and Formatting: Major English correction is necessary. The manuscript content is interesting, but the English must be revised prior to the resubmission else it will not be further considered. Please find a native speaker/proofreading company to crosscheck the English.1.     Some of the examples are as follows:

Line 39: E.g., ... We do not use E.g., like this in a written manuscript. Better phrases should be "for instance

Conclusion section seems to be a repetition of the results section. Huge modifications are required. Please make sure your ‘conclusion’ section underscores the scientific value added to your paper, and/or the applicability of your findings/results, as indicated previously. Please revise your conclusion part into more detail. Basically, you should enhance your contributions, and limitations, underscore the scientific value added to your paper, and/or the applicability of your findings/results and future study in this section.

Implications for future research may also be included in the conclusion at the end.

Some relevant references might be missing in the manuscript.

(i) https://doi.org/10.1016/j.jclepro.2022.132893

(ii) https://doi.org/10.1007/s40808-021-01107-6

Author Response

Response to Reviewer 3 Comments

The authors thank the reviewer for the time spent to comment and improve the manuscript. We have revised the manuscript accordingly, taking the comments from the reviewer into consideration. Below are answers to the comments and description of actions taken: The black text is comments from the reviewer, and the Red text is our response.

Point 1: Abstract: Too general and I cannot identify the research gap that you are addressing. Please revise.

Thank you for your comment. The abstract has been revised now and looks as follow: (lines 15-31) “Air quality issues still affect the quality of life for people in industrialised cities around the world. The investigations should include the identification of the sources of the pollution and its distribution in space and time. This work is the first attempt to perform identification of the sources of pollution in Ust-Kamenogorsk city in Kazakhstan. Analysis of retrospective data (including ten variables (TSP, SO2, CO, NO2, phenol, HF, HCl, H2SO4, formaldehyde, H2S) from five monitoring stations for the period 2017-2021) using multivariate statistical methods and hierarchical cluster analysis has been performed to assess spatiotemporal patterns of air quality of the city. The results indicate that the contamination patterns can be grouped into two categories: cold and warm seasons. The study revealed the dangerous concentrations of NO2 and SO2 exceeded the limits by 2-3 and 1.5-2 times, independently of the seasonality. Averaged concentrations of TSP slightly exceeded the established limits for the most industrialised part of the city. Concentrations of HF and formaldehyde significantly rose during the cold seasons compared to the warm seasons. Other chemical parameters significantly depend on the seasonality and locations of the sampling points. The major reason for air pollution is twofold – the use of a burnt-coal throughout the year for electricity and heat generation (especially during the cold seasons) and the high density of the heavy metallurgy industry in the city. The principal component analysis confirms a high loading of industrial sources of air pollution on both spatial and seasonal dimensions.”.

Point 2: Introduction: Major revision is required. The research gap and the research objectives were not clear in the submission. Line 92-93: Not professionally written. Please provide more information. Provide a more detailed literature review about the conditions and highlight what is your contributions.

Thank you for your comment. The Introduction has been revised accordingly and now looks as follow: (lines 64-78) “One of the problematic cities in the country, Ust-Kamenogorsk, is located in a major mining and smelting area and has been rated among the worst cities regarding the problems of ecology according to World Bank [14]. Extreme pollution events from accidental emissions are not rare in the city [15]. Descriptive studies of poor air quality in the city have said that the city has been rated the most SO2-polluted city in Kazakhstan and among the five most-polluted NO2 and O3 cities in Kazakhstan from 2011 to 2017 [12]. A study about the impact of COVID-19 lockdown on air quality in the city has revealed that the level of CO has decreased by 21–23 % with the increase of the TSP level by 13–21 %, and had no significant effect on SO2 and NO2 concentrations in the city [16]. Elevated levels of trace elements, particularly Ba, Mn, Pb, V, and Zn, in the blood of city residents of Ust-Kamenogorsk have been found [17], which could indicate a severe impact on the industrial activities of the city. The known studies are limited by their descriptive origin, while there is a demand for deep investigations of the relationships between a high level of air pollution, weak environmental regulation, pyrometallurgical processes and dependence on coal-burnt energy.”

Lines (92-130): “According to an opinion from [24], preliminary investigations play a core role in the identification of the sources of pollution to drive “the wheel of progress: Sources ⇒ Effects ⇒ Regulation ⇒ Control”. There are known different ways to investigate potential sources apportionment: emission inventories [25], inverse modelling [26], artificial neural networks [27], receptor modelling methods [28], air quality models [25], a combination of the different approaches and models [29], application of positive matrix factorisation [30], and even chemical analysis of biomonitoring species [31]. However, all of the listed methods demand a very detailed dataset with a number of monitored parameters. This limitation is a serious obstacle in the conditions of many developing countries, including Kazakhstan, where the system of permanent monitoring with extended parameters, such as PMs, has still been established [32].

Under these conditions, one of the efficient ways to investigate sources apportionment with the following understanding of the needed steps for air quality management can be multivariate statistical techniques, particularly the Principal Component Analysis (PCA) and the Cluster Analysis. Also, using analysis of scores and loadings in PCA can give a representation of potential chemical reactions in ambient air [33]. The approach to applying PCA for the assessment of patterns in air quality has already been applied several times. For instance, Dominick et al. [34] have aimed to investigate possible sources of air pollutants and spatial patterns in Malaysia using the same techniques as the authors plan to do with grouping particulate contaminants in principal components. Application of the PCA to find the correlation between gaseous pollutant concentrations, meteorological factors and potential sources of pollution has identified the contribution of combustion- and non-combustion-related emitters in Greece [35] or in India [36]. The tool has been used for the same purpose also by Azid et al. in the context of the prediction of air pollution [37]. PCA has supported the assessment and identification of the sources of air pollution in India, where complex industrial activities exist [38]. Revealing information about the sources and mechanisms of air pollution in Madrid and visualising their spatial distribution using PCA and the geostatistical method has been carried out by Núñez-Alonso et al. [39]. Particular attention has been paid to the investigation of the presence of metals only in particulate matters coupled with multivariate statistical techniques in an Iranian industrial city [40]. For example, a combination of the analysis of the chemical composition of fine particulate matters with the focus on metals presence with Absolute Principal Component Analysis let to explain identified pollutants to their sources in the USA [41].

The aim of this study is to analyse key factors impacting air quality in Ust-Kamenogorsk using the available dataset for the period 2017-2021 by multivariate statistical techniques on spatial and temporal scales. This approach lets to investigate potential sources of apportionment using the large but limited, a dataset of the observations in the city for the first time.”

Point 3: Study area: Fig.1 better naming should be provided. Also, provide the scale bar on the map.

Thank you. The Figure has been modified.

Point 4: Results and Discussions: Fig.2: Very hard to read. Suggest making two rows for better illustration. Fig 3: Similar issues: Poor arrangement. Please find a better arrangement method. Perhaps Table 2 can be better illustrated using box plots.

Thank you for your comments. In general, we have met several challenges while had been presenting our results. Regarding Table 2 and Figure 3: they complement each other. From the one hand, we aimed to present the exact values, including such indicators, as mean and median values. They are better to be shown in a numeric form. From the other hand, to highlight such high concentrations on the seasonal scale was better using a format on Figure 3. Another challenge is to present information about all ten contamninants, which assumes making some complicated graphs.

Regarding Figure 2: we did our best to explain it inside the text during the sub-sections 2.3. Data management and methodological framework and 3.1. Hierarchical clustering analysis with the reference to Appendix A, where the presentation of the hierrarchical cluster analysis has shown in its regular form of dendrograms for each observation station.

Point 5: Good discussion in section 3.3

Thank you for this point.

Point 6: Table 4: Consider coloring the bolded value.

Thank you. Done.

Point 7: Fig 4: Again, poor arrangement and low resolution of the figures.

Thank you for this comment. The resolution is 300 dpi, which satisfies the requirements of the journal. Also, the purpose of the pictures is to show the spatial distribution of the pollution. We guess that quality of the picture corresponds this purpose.

Point 8: There should be a separate section for discussion of findings. Further, findings must be compared with previous researchers’ work. Unique contributions of research should be highlighted.

Thank you for this note. While the section is titled Results and Discussion, we have tried to present disscussion of the findings through the presentation of the results for each sub-section. However, your suggestions have been considered, which has been reflected in the text as follows: (lines 387-401) “Table 4 summarises a conditional grouping of identified PCs, results of the spatiotemporal assessment, hierarchical clustering analysis, and respective chemicals. Group 1 for the cold seasons can be explained by the intensive use of the fossil fuel-fired by all users: power plants, industry, and households coupled with meteorological conditions (based on Figure 5). This mix ensures a variety of pollutants are released into the atmosphere by the burning of different types of fuel. The same group for the warm seasons indicates and highlights the problem of coal consumption, mainly by industrial enterprises, which is indicated by the uniform seasonal distribution of NO2 and SO2 [4]. Group 2 for the cold seasons can be explained by specific industrial processes and the release of the associated contaminants into the atmosphere. Group 2 for the warm seasons could indicate a contribution of traffic to air pollution. While local authorities have attempted to explain air quality issues by a high density of motor transport, the results of this study confirm outcomes of previous research that heavy air pollution in the city caused by non-transport-related sources [67] that this factor has a low significance comparatively with the impact of industrial activities, especially burnt-coal based.”

Points 9-10: Grammatical and Formatting: Major English correction is necessary. The manuscript content is interesting, but the English must be revised prior to the resubmission else it will not be further considered. Please find a native speaker/proofreading company to crosscheck the English.

Thank you for your suggestion. The text has been edited and revised.

Points 11-12: Conclusion section seems to be a repetition of the results section. Huge modifications are required. Please make sure your ‘conclusion’ section underscores the scientific value added to your paper, and/or the applicability of your findings/results, as indicated previously. Please revise your conclusion part into more detail. Basically, you should enhance your contributions, and limitations, underscore the scientific value added to your paper, and/or the applicability of your findings/results and future study in this section.

Implications for future research may also be included in the conclusion at the end.

Thank you for this valuable suggestion. The Conclusion has been revised and now looks as follows: lines (403-432) “Ust-Kamenogorsk is one of the most important industrial centres of Kazakhstan, where non-ferrous metals are produced are exported abroad for the largest world companies. Active consumption of coal and raw material orientation of the industrial enterprises make Ust-Kamenogorsk one of the most polluted industrial cities in the world. This study, for the first time, aimed to analyse spatiotemporal patterns of air pollution in the city and to identify potential sources of apportionment using multivariate statistical techniques. The results show that the combination of large enterprises and coal-fired thermal power plants severely affects air quality in Ust-Kamenogorsk. The average concentrations of SO2 and NO2 for the entire study period exceeded the standards of WHO and Kazakhstan within the whole city all year round. The major emitters are located in the northern industrial zone with two huge metallurgy factories and one thermal power plant. The principal component analysis revealed that emissions of TSP, SO2, CO, H2S, NO2, HF, and H2SO4 with high likelihood are of industrial origin, which is significantly aggravated during the cold seasons. While official reports show a decrease in industrial emissions, air quality has not been improved even during the warm seasons when households stop heating with the expected elimination of air pollution.

The results of this study can be valuable for decision-makers in developing and applying respective strategies and actions for a decrease in air pollution and elimination of the social and environmental effects. The main limitation of the research is the difficulty in determining the ratio of the contributions of the major emitters to the high level of air pollution. Future research will focus on a detailed investigation of the composition of unstudied contaminants (particularly, particulate matters) in both ambient air and the zones of industrial enterprises. This research should be combined with the modelling of source profiles or fingerprints.

The air quality issues in Ust-Kamenogorsk most probably have been caused by a combination of the factors: weak environmental regulation and control, the influence of large companies on the legislative process through large professional associations, and an outdated and energy-intensive industry. The authors hope that the outcomes of this and other studies would be a sufficient research basis for authorities to develop the right strategy for air management in the region.”.

Point 13: Some relevant references might be missing in the manuscript.

Thank you. The references have been used and inserted.

Round 2

Reviewer 3 Report

Comments to the authors:

The authors have substantially addressed my comments, however, still there are some points to be addressed:

(i) Legend for Fig 4a and b for 10 parameters are too small. Enlarge it. 

(ii) Abbreviation for Figure 5.